# Study on Varicella-Zoster Virus Antibody Levels among Children Aged 1–7 Years in Changzhou, China

**DOI:** 10.3390/vaccines12030290

**Published:** 2024-03-11

**Authors:** Dan Wu, Changlei Han, Suting Xiong, Peipei Zhang, Han Gao, Junhong Li, Fengming Wang, Qinwen Xu, Xin Dong

**Affiliations:** 1Changzhou Center for Disease Control and Prevention, Changzhou 213022, China; iron_wd999@163.com (D.W.); hanlei_2001@163.com (C.H.); 1105609411xst@sina.com (S.X.); zhangpeipei0504@126.com (P.Z.); gaohan_mcrsft@outlook.com (H.G.); lijunhong@njmu.edu.cn (J.L.); wfm0519@163.com (F.W.); 2Changzhou Institute for Advanced Study of Public Health, Nanjing Medical University, Changzhou 213022, China; 3Changzhou Center for Disease Control and Prevention, Zhonglou District, Changzhou 213022, China; xqw168@126.com

**Keywords:** varicella, vaccination, VZV-IgG, GAMLSS

## Abstract

We aim to understand the varicella-zoster virus (VZV) antibody levels in children after vaccination and to construct VZV-IgG centile curves and reference values for children aged 1–7 years. From September to October 2023, a total of 806 children were recruited according to the time intervals of 1 month, 6 months, 1 year, 2 years, and 3 years after vaccination, as well as age groups. A generalized additive model for location, shape, and scale (GAMLSS) was applied to estimate P_3_, P_10_, P_25_, P_50_, P_75_, P_90_, and P_97_ centile reference values of VZV-IgG, and 95% reference intervals were calculated. A total of 785 children were included in the analysis, with an overall positivity rate of 70.3%, a median antibody concentration of 192.05 (82.89–571.14) mIU/mL, and a positivity rate of 57.7% for one dose of vaccine and 84.2% for two doses. Antibody positivity rates at 1 month, 6 months, 1 year, 2 years, and 3 years after vaccination were 65.1%, 74.4%, 80.4%, 67.7%, and 63.0%, respectively. The GAMLSS results showed that VZV-IgG had a tendency to increase and then decrease after vaccination, and the second dose of vaccination could significantly increase VZV-IgG. Two doses of varicella vaccine should be administered to children in a timely manner and included in the routine vaccination programs.

## 1. Introduction

Varicella is a common infectious disease caused by the human herpesvirus—varicella-zoster virus (VZV)—mainly transmitted through respiratory droplets or direct contact with the varicella or herpes zoster rash [1]. Varicella is highly contagious, children aged below 14 years are more susceptible to VZV, which usually causes outbreaks in collective institutions such as childcare and primary and secondary schools. Varicella has been at the top of the list of public health emergencies of infectious diseases in Changzhou for many years. Moreover, following the primary infection, VZV becomes latent in neurons in the peripheral ganglia and can be reactivated several years later, thus causing herpes zoster and neurological dysfunction [2,3,4]. The social and economic burden caused by varicella is severe, and vaccination is the most successful and cost-effective intervention to prevent it.

The safety and efficacy of varicella vaccines (VarV) have been demonstrated since the development of the live attenuated varicella vaccine (vOka) by Michiaki Takahashi in 1974 [5], which can effectively control varicella incidence and outbreaks [6]. Nonetheless, when VarV is included in routine childhood immunization programs, there are differences in vaccination strategies in different regions. For example, in the United States, the Advisory Committee on Immunization Practices (ACIP) approved a routine two-dose VarV schedule for children in 2006 [7]; the Government of Japan changed the VarV schedule from voluntary to routine vaccination in October 2014, in conjunction with the introduction of a two-dose VarV schedule [8]. There are still countries such as Australia, New Zealand, Argentina, and Oman that recommend a one-dose VarV schedule [9]. In addition, although several countries have adopted a two-dose VarV schedule, the timing of the second dose of vaccine varies. Japan requires two doses of vaccine by 3 years of age (with an interval of at least 3 months) [8]; the United States requires a first dose of VarV at age 12–15 months and the second at 4–6 years [7,10]; and Germany recommends the first dose at 11–14 months and the second dose (with an interval of at least 1 month) at 15–23 months [9].

Currently, VarV has not been included in the national immunization program (NIP) in China. Cities such as Beijing, Hangzhou, and Suzhou have gradually incorporated VarV into their local routine vaccination programs. At present, Changzhou is implementing the free strategy of two doses of VarV, with the first dose at 12–18 months and the second dose at 4 years, with an interval of at least 3 years between vaccinations. Timely surveillance of varicella antibody levels in school-age children helps to identify susceptible populations, scientifically predict the epidemiological trend of varicella, and evaluate the effect of immunization strategy implementation. In this study, we examined the concentration of VZV antibodies in different doses of VarV and used generalized additive models for location, scale, and shape (GAMLSS) to draw centile curves based on the VZV antibody levels of children aged 1–7 years in Changzhou. We aim to provide scientific evidence for local administrations on the development and optimization of VZV immunization strategies that can serve as references for other regions of the world.

## 2. Materials and Methods

### 2.1. Study Design

A cross-sectional survey was conducted between September and October 2023 in Changzhou. Subjects were divided into two age groups, including the “1-year-old group” and “4-year-old group “, and into five time levels, including 1 month, 6 months, 1 year, 2 years, and 3 years, according to the time interval after the latest VarV, for a total of 10 subgroups. Age groups were determined using a backward-looking approach. For example, if the sampling date was set at 15 September 2023–30 September 2023, the 1-month interval after the latest VarV was set at 1 August 2023–30 August 2023, and the “1-year-old group” was defined as those born between 30 August 2021 and 30 August 2022, and the “4-year-old group” was defined as those born between 30 August 2018 and 30 August 2019. The remaining groups were defined in a similar manner.

Subjects were recruited based on the Vaccination Integrated Service Management Information System of Jiangsu Province, China. The system is a standardized, computerized database that collects vaccination information, including name, gender, date of birth, vaccine name and manufacturer, date and institution of vaccination, and other relevant vaccination data. A systematic sampling method was adopted for recruitment, and if the subjects refused to cooperate with the survey, they were excluded, and eligible subjects were selected sequentially. The survey was stopped once the sample size was met for each subgroup. Subjects in each subgroup had venous blood collected and completed questionnaires. Due to the principle of data confidentiality, strict training was given to the staff beforehand to avoid data leakage. Moreover, each subject was given a unique ID number when filling out the questionnaire, the questionnaire completion and the varicella antibody laboratory test were independent, and only the ID number was retained on the sampling tubes without any other information.

### 2.2. Vaccine

A live attenuated varicella vaccine (Oka strain) was recommended for local routine childhood immunization in Changzhou and provided by 3 domestic vaccine companies (Shanghai Institute of Biological Products Co., Ltd., Shanghai, China, Sinovac (Dalian) Vaccine Technology Co., Ltd., Dalian, China and Changchun BCHT Biotechnology Co., Ltd., Changchun, China). The vaccines were derived from the original Japanese vOka strain in the 1970s and all called “Varicella Vaccine, Live”. The GenBank number for the first two companies was AB097932.1, and the last company was AB097932 (obtained by sequencing the virus of a random vOka stock used).

### 2.3. Ethics Statement

Written informed consent was obtained from all participants’ guardians prior to this study. This study was conducted in accordance with the Declaration of Helsinki, and the protocol was approved by the Ethics Committee of Changzhou Institute for Advanced Study of Public Health, Nanjing Medical University (Ethical Approval code: 202303).

### 2.4. Sample Size Estimation

Based on the sample size formula for the status quo survey [11]:$$n=\frac{t_{\alpha }^{2}}{e^{2}}p\left(1-p\right)$$
where *α* = 95%, *e* = 5%, and *p* = 0.57 [12], and taking into account a certain dropout rate, we calculated that 400 subjects were needed for each age group. A total of 800 venous blood collections were required for the 1-year-old group and 4-year-old group.

### 2.5. Inclusion and Exclusion Criteria

The inclusion criteria included the following: (1) children aged 0–7 years; (2) had a clear history of VarV; (3) had no organic diseases or psycho-neurological diseases; and (4) obtained informed consent from guardians.

The exclusion criteria included the following: (1) those suffering from acute diseases, severe chronic diseases, acute exacerbation of chronic diseases, and fever; (2) those with immunodeficiency, immunocompromised, or under immunosuppressive treatment; (3) those who had acute diseases, infections requiring systemic antibiotics, or antiviral treatment in the past 7 days; and (4) any conditions that might affect the evaluation of the test in the opinion of the observers.

### 2.6. Measures

Questionnaire survey: The questionnaire survey included information on the name, sex, date of birth, household registration, VarV history, and history of varicella of the study subjects.

Specimen collection and VZV-IgG detection: A total of 3–5 mL of venous blood was collected from each subject, serum was separated within 6 h after blood specimen collection, and venous blood specimens without centrifugation were not frozen to avoid repeated thawing hemolysis. Serum was stored at −20 °C after separation and promptly sent to the laboratory of Changzhou Center for Disease Control and Prevention (CDC) for detection. VZV-specific antibodies were measured with a quantitative ELISA method, which detected antibodies against VZV glycoproteins prepared from human fibroblasts infected by VZV. Antibody detection was performed using the VZV-IgG antibody detection reagents (Product Number: ESR104G) produced by Virion/Serion GmbH Institute (Würzburg, Germany). According to the manufacturer’s instructions, the quantitative detection range of VZV-IgG is 15–2000 mIU/mL, where VZV-IgG >100 mIU/mL or <50 mIU/mL are determined as positive and negative, respectively, and 50–100 mIU/mL is defined as equivocal. In our study, VZV-IgG > 2000 mIU/mL or <15 mIU/mL were calculated as 2000 mIU/mL and 15 mIU/mL, respectively. Equivocal samples were reviewed using the same batch of kits, and if the results were equivocal, they were determined to be negative.

### 2.7. Statistical Analysis

EpiData version 3.1 (EpiData Association, Odense, Denmark) statistical software was used for data double entry and verification, and all statistical analyses were conduct using R version 4.3.2. Categorical variables were presented as frequencies and percentages, and differences were analyzed using the χ^2^ test. Continuous variables were presented as mean and standard deviation according to the normality test for those obeying normal distribution, and Student’s *t*-test was used for the difference analysis, and median (P_25_–P_75_) for those not obeying normal distribution, and the Wilcoxon rank-sum test was used for the difference analysis.

A GAMLSS was used to estimate VZV-IgG reference values and draw centile charts [13,14,15]. The GAMLSS requires a parametric distribution assumption for the response variable and models the distribution function of the explanatory variables using a nonparametric smoothing function, making the GAMLSS a semi-parametric regression type model [16] with the formula shown below:y~D(μ,σ,υ,τ),
where μ represents the mean, σ represents the variance, υ represents the skewness, and τ represents the kurtosis. The first two parameters are characterized as location and scale parameters, and the remaining parameters are characterized as shape parameters. In this study, we fitted data to the Box–Cox-t (BCT) distribution, D=(μ,σ,υ,τ), and the Box–Cox Cole and Green (BCCG) distribution, D=(μ,σ,υ), respectively, by taking the logarithm of VZV-IgG as the dependent variable and the corresponding age as the explanatory variable. Each parameter was smoothed using a cubic spline or *p*-spline smoothing function [17,18,19,20,21]. Model fitting was evaluated using a combination of Akaike information criterion (AIC) and Schwarz Bayesian criterion (SBC). Finally, the P3, P10, P25, P50, P75, P90, and P97 centile reference values of VZV-IgG among children aged 1–7 years in Changzhou were constructed according to the optimal model, and the centile chart was plotted.

## 3. Results

### 3.1. Basic Information

A total of 806 subjects were recruited. All of them underwent informed consent and varicella antibody testing and were included in the GAMLSS construction. In the analysis of VZV-IgG differences by groups such as age and sex, 21 were excluded after verifying that they did not meet the time interval (see Figure 1 for details). The average age of the 785 subjects was 4.00 ± 1.928 years, with 383 in the 1-year-old group and 402 in the 4-year-old group; in terms of sex, there were 397 males and 388 females; and the median antibody concentration of VZV-IgG was 192.05 (82.89–571.14) mIU/mL. The number of positives totaled 552, accounting for 70.3%. In terms of VarV history, 411 people received one dose of vaccine and 374 people received two doses of vaccine; all subjects in this study reported never having varicella.

### 3.2. Univariate Analysis of VZV-IgG

A comparison of VZV-IgG seropositivity rates with different characteristics is shown in Table 1. Among the overall study population, the antibody positivity rate was 57.7% for the one-dose Varv and 84.2% for the two-dose VarV, with significant differences between doses (χ^2^ = 66.184, *p* < 0.05). The antibody positivity rates were 58% and 82.1% in the 1-year-old and 4-year-old groups, respectively, and were significantly higher in the 4-year-old group (χ^2^ = 54.698, *p* < 0.05). In terms of the time interval after vaccination, the positivity rate was 65.1% for the interval of 1 month, 74.4% for the interval of 6 months, 80.4% for the interval of 1 year, 67.7% for the interval of 2 years, and 63.0% for the interval of 3 years (χ^2^ = 15.375, *p* < 0.05). The positivity rate demonstrated a trend of increasing and then decreasing, reaching the highest level at 1 year of postvaccination, and then slowly decreasing and falling to the lowest level at 3 years of postvaccination. No statistically significant difference in the VZV-IgG seropositivity rates by gender and household registration was found (*p* > 0.05)

We conducted stratified analysis sequentially on the age groups, VarV doses, and time intervals. Statistically significant differences in VZV-IgG seropositivity rates at different time intervals after vaccination were observed in the 1- and 4-year-old groups, and in the one- and two-dose VarV groups (*p* < 0.05). Figure 2 shows the VZV-IgG seropositivity rate of different VarV dose groups, and it can be seen that in the one-dose VarV group, the positivity rate reached the highest (75%) at the first year of postvaccination and then slowly decreased; in the two-dose VarV group, the positivity rate firstly peaked at 1 month of postvaccination (95.2%), then declined to 6 months of postvaccination, increased again until 1 year of postvaccination, and finally, declined slowly. Meanwhile, the stratified analysis by time interval after vaccination revealed that the positivity rate of the two-dose group was significantly higher than that of the one-dose group at all time intervals, except for the 1 year after vaccination (*p* < 0.05).

### 3.3. GAMLSS

#### 3.3.1. Model Construction and Fitting

The BCT distribution and BCCG distribution were fitted sequentially after logarithmic treatment for antibody levels, while different smoothing functions were applied to each parameter, and the results are shown in Table 2. In the case of convergence of the parameter estimation, as well as according to the SBC, AIC minimization principle, M16 was the optimal model, i.e., a cubic spline function smoothing for *υ*, a *p*-spline function smoothing for *μ* and *σ*, and fitting the BCCG distribution to the antibody level. *μ*, *σ*, and *υ* were not constant across ages. In Figure 3a–c refer to the variations with age for μ, σ, and υ, respectively, which also illustrated the applicability and superiority of the GAMLSS.

#### 3.3.2. Centile Reference Values and Curves

The GAMLSS method was used to construct centile reference values (Table 3) and curves (Figure 4) for P_3_, P_10_, P_25_, P_50_, P_75_, P_90_, and P_97_ of VZV-IgG at different ages. The curves were generally smooth and fitted well, and it can be seen from Figure 3 that the antibody levels of children aged 1–4 years and 4–7 years both showed a trend of increasing and then decreasing. The antibody levels increased after the first dose of vaccine at the age of 1 year and remained stable or declined slightly until the age of 3 years, climbed significantly after the second dose of vaccine at the age of 4 years, and then gradually decreased with increasing age. According to the centile reference values, children aged 1–3 years only reached 2 (≥2 was considered to have a protective effect) at P_50_, while children aged 4–7 years already reached 2 at P_25_. We also calculated the 95% reference interval for each age group (see Table 3 for details).

## 4. Discussion

There are numerous methods for developing reference intervals, such as the percentile method, the normal distribution method, and the Hoffmann method. These methods are simple and easy to use for skewed and normally distributed data, but they are not smooth enough when plotting centile curves to cope with the parameter differences in nonlinear parameters at different ages. The GAMLSS, as a progression and extension of the LMS method, is the standard method recommended by the World Health Organization (WHO) for plotting growth curves. The GAMLSS takes into account the location parameter (μ), the scale parameter (σ), and the shape parameters (υ, τ) while combining various distributions such as BCCG and BCT to make the fitted values closer to the real values. There are limited studies on the establishment of reference values for VZV antibody levels, and the results of this study can help identify trends in VZV antibodies in children aged 1–7 years under a two-dose vaccination strategy, as well as provide reference values for antibody levels in children of different ages. According to the centile curves, it is clear that under the implementation of the two-dose VarV immunization strategy, the majority of children reached the protective level of antibodies (≥2), and after the first dose of VarV, the antibody levels in children (1–3 years old) increased and then decreased, with nearly 50% of children not receiving effective protection. In contrast, after the second dose of VarV, the antibody levels in children (4–7 years old) were significantly elevated, with greater than 75% of children producing protective antibody levels, suggesting the need for implementation of a second dose of VarV to help provide sufficient immunity to protect children from varicella infection. In addition, we provided centile reference values based on a GAMLSS, which can help determine the percentile position of varicella antibody levels in children and the possible future trajectory of change. Thus, guardians can target more non-pharmacological interventions, such as wearing face masks and washing hands frequently, to children in the low percentile range during the varicella epidemic season. In the future, we will consider including more children in our study to further observe antibody levels at 5 or 10 years of postvaccination.

In recent years in China, varicella has ranked first among vaccine-preventable infectious diseases, with an average annual incidence of 55.05/100,000 in 2016–2019 [22,23], resulting in a considerable disease burden with huge healthcare and social costs. VarV was not included in routine childhood immunization programs in Changzhou before 2020, and children received VarV at their own expense and on a voluntary basis. However, the coverage of the varicella vaccine was low, far from the 80% vaccine coverage rate recommended by WHO [24], and thus was unable to constitute an effective immunization barrier. According to varicella surveillance data captured by the China Information System for Disease Control and Prevention, the annual incidence rates of varicella for the years 2017–2019 were 85.95, 190.76, and 259.63/100,000, respectively. Since 2020, Changzhou has implemented a free two-dose VarV strategy, and the annual incidence rates of varicella were 124.81, 86.64, 62.99, and 66.92 per 100,000 from 2020 to 2023, respectively. The average annual incidence rate in Changzhou has decreased from 178.781/100,000 in 2017–2019 (pre-implementation) to 85.34/100,000 in 2020–2023 (post-implementation). From a macroscopic point of view, the current two-dose VarV strategy can effectively reduce the incidence rate of varicella and prevent varicella outbreaks.

The VZV antibody is an important indicator for the evaluation of the immunity effect after VZV infection or vaccination, which helps to understand the seroepidemiology of VZV in children and to assess the immunity effect of immunization with VarV in different doses and different age groups. The results of this study showed that the overall positivity rate of varicella-vaccinated children aged 1–7 years in Changzhou was 70.3%, with a one-dose positivity rate of 57.7% and a two-dose antibody positivity rate of 84.2%. Both one and two doses of VarV provided considerable protection to children (>50% seropositivity rate). A survey of varicella antibody levels in 1772 healthy children aged 0–11 years in Sichuan Province, China, in 2022, showed that among the unvaccinated population, the VZV-IgG seropositivity rate was 8.97% for 1–2 years old and 18.75% for 3–5 years old [25]. These values were lower than the seropositivity rate of the current study subjects and indirectly proved the benefits of VarV, which can play a crucial role in preventing varicella. Meanwhile, our findings are close to the results of existing reports in China, such as 67.62% of seropositivity among children aged 1–14 years with vaccination history in Hangzhou, China [26], 63.6% of seropositivity among 555 students in grades 1–4 with vaccination history in Taiwan [27], and 89.1% in the population with a history of two-dose varicella immunization in Harbin [28]. Furthermore, in combination with the change trend, the antibody positivity rate and antibody level reached the highest after 1 year of varicella vaccination, regardless of the one-dose or two-dose group. When exceeding the vaccination interval of 1 year, they all showed a trend of decreasing with the increase in the time interval, especially for the one-dose group. The positivity rate 3 years after vaccination was only about 50%, which could not protect against varicella infections effectively. Therefore, breakthrough cases may appear, which also suggests the need for a second dose of VarV. Surveillance data on varicella outbreaks in Shanghai from 2008 to 2014 showed that despite the implementation of one dose of Varv in Shanghai, varicella outbreaks in kindergartens and schools increased and that one dose of VarV could not prevent varicella outbreaks in kindergartens and schools [29]. In addition, routine VarV has been available in the United States since 1995, and Chaves conducted a 10-year active surveillance (1995–2004) covering 350,000 study participants, which showed that 11,356 (9.5%) developed breakthrough varicella. Furthermore, the annual incidence of breakthrough varicella increased significantly with time after vaccination, from 1.6 cases/1000 person-years at 5 years to 9 cases/1000 person-years at 5 years, increasing to 58.2 cases/1000 person-years at 9 years, suggesting that two doses of varicella vaccine were needed [30]. It is worth noting that although live vaccines generally provide long-lasting immunity, a decline in varicella antibody level titers is inevitable. To date, only a few studies have shown that VarV provides protection for nearly 10 years after vaccination. An Italian retrospective cohort study showed that VZV antibody levels tend to decline within 1 year of completing the primary VarV, and after 9 years, half of the fully vaccinated population had lost circulating antibodies [31]. VZV antibody titers may decline further if the titers are measured in young adults from China who were immunized during childhood. Therefore, the necessity of administering a third booster dose of VarV for particular populations is worth considering.

Similarly, our research findings also indicated that the two-dose vaccination strategy was effective, with a stronger protective effect compared to the one-dose vaccination [32,33], with a nearly 30% higher positivity rate in the overall positivity rate in the two-dose than in the one-dose. In terms of the persistence of antibodies, especially at 2 and 3 years after the vaccination, the two-dose group demonstrated a higher positivity rate, and its decline was much lower. After the age of 4, children’s exposure to the outside world increases due to school attendance, and consequently, the chances of contracting VZV naturally increase significantly, and two doses of VarV would allow for at least 75% of children aged 4–7 years to have a protective level of antibodies. Furthermore, although the second dose of VarV has been shown to increase effectiveness significantly, the time between the two doses of vaccination is debated [34], e.g., short-interval two-dose varicella immunization schedules have been used in Germany [9], whereas long-interval varicella vaccination has been used in China and the U.S. Our results provided evidence for a long-interval vaccination strategy, with no breakthrough cases in the population, and suggested that two doses of vaccination had fair antibody persistence and ensured high protection until 7 years of age. In conclusion, a second dose of varicella vaccine is needed, and long intervals of varicella vaccination are feasible to improve protection against failure of the first dose and vaccine-induced immune decline.

There are limitations in this study. Firstly, this study was limited to the Changzhou area and did not involve children over 7 years old. Subsequent studies may consider expanding the study population so that the results can be extrapolated more easily. Secondly, this study was a cross-sectional investigation, and further cohort populations should be established to monitor the change in antibody levels with age in the same population. Finally, during the construction of the GAMLSS, we also attempted to fit the BCPE distribution for the VZV-IgG, and although the AIC and the SBC were much lower, the centile curves showed overfitting, and we chose the BCT distribution for the fitting in the end.

## 5. Conclusions

In view of the current high incidence of varicella, relevant health authorities should consider incorporating VarV into the national immunization program and recommend a two-dose immunization strategy for better control of varicella transmission.

## Figures and Tables

**Figure 1 vaccines-12-00290-f001:**
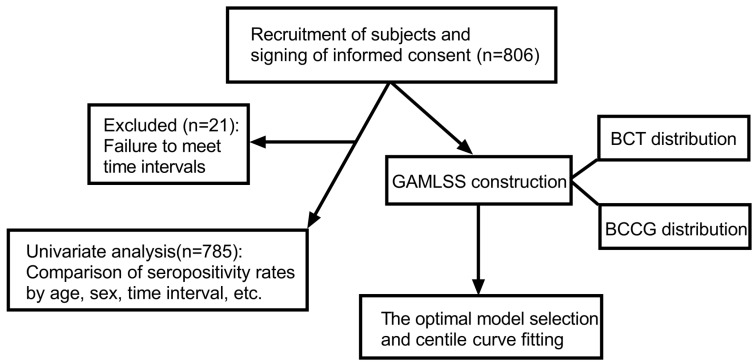
Research flowchart.

**Figure 2 vaccines-12-00290-f002:**
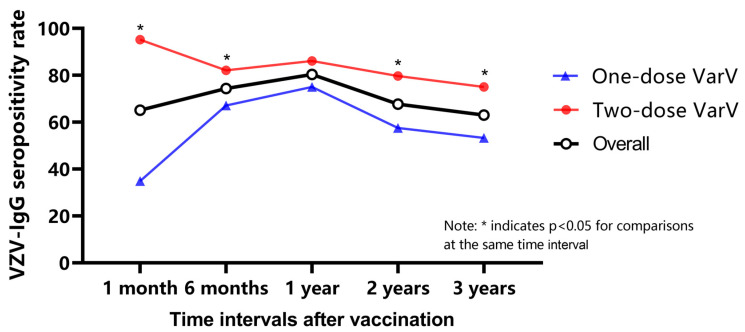
Comparison of seropositivity rates for different doses at each time interval.

**Figure 3 vaccines-12-00290-f003:**
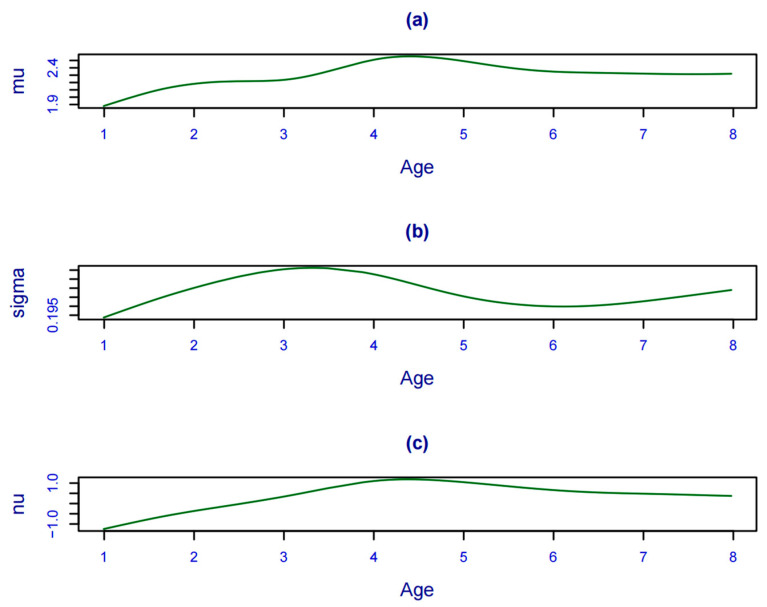
Variation in each parameter with age. (**a**–**c**) refer to the variations with age for μ, σ, and υ, respectively.

**Figure 4 vaccines-12-00290-f004:**
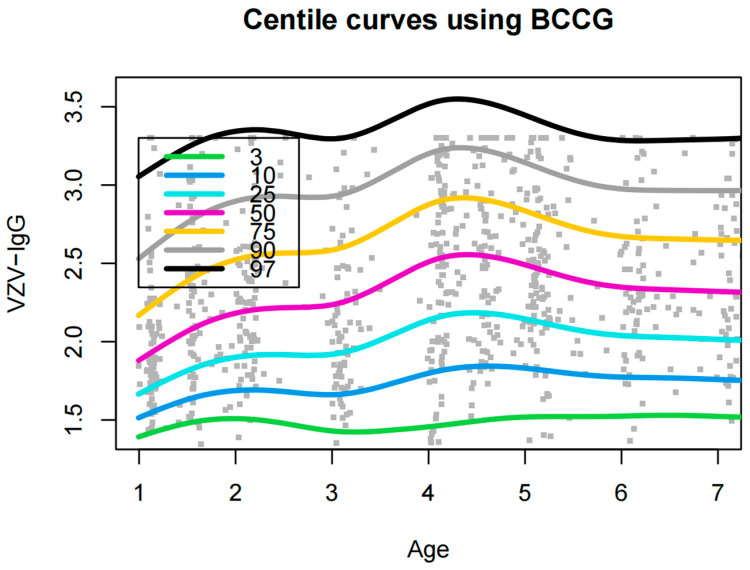
Centile curves for VZV-IgG (logarithmic) in children aged 1–7 years.

**Table 1 vaccines-12-00290-t001:** Comparison of VZV-IgG seropositivity rates with different characteristics.

Characteristic	Positive (n = 552)	Negative (n = 233)	χ^2^	*p*
Overall analysis				
Dose			66.184	0.000
One dose	237 (57.7%)	174 (42.3%)		
Two doses	315 (84.2%)	59 (15.8%)		
Age group			54.698	0.000
1-year-old group	222 (58.0%)	161 (42.0%)		
4-year-old group	330 (82.1%)	72 (17.9%)		
Sex			0.244	0.621
Male	276 (69.5%)	121 (30.5%)		
Female	276 (71.1%)	112 (28.9%)		
Household registration			0.248	0.618
Permanent population (residence < 3 months)	402 (69.8%)	174 (30.2%)		
Floating population (residence ≥ 3 months)	149 (71.6%)	59 (28.4%)		
Time intervals after vaccination			15.375	0.004
1 month	108 (65.1%)	58 (34.9%)		
6 months	119 (74.4%)	41 (25.6%)		
1 year	131 (80.4%)	32 (19.6%)		
2 years	109 (67.7%)	52 (32.3%)		
3 years	85 (63.0%)	50 (37.0%)		
Stratified analysis				
Stratified by age group				
1-year-old group			32.920	0.000
1 month	28 (34.6%)	53 (65.4%)		
6 months	54 (67.5%)	26 (32.5%)		
1 year	61 (76.3%)	19 (23.7%)		
2 years	47 (58.0%)	34 (42.0%)		
3 years	32 (52.5%)	29 (47.5%)		
4-year-old group			15.349	0.004
1 month	80 (94.1%)	5 (5.9%)		
6 months	65 (81.2%)	15 (18.8%)		
1 year	70 (84.3%)	13 (15.7%)		
2 years	62 (77.5%)	18 (22.5%)		
3 years	53 (71.6%)	21 (28.4%)		
Stratified by dose				
One dose			31.449	0.000
1 month	29 (34.9%)	54 (65.1%)		
6 months	55 (67.1%)	27 (32.9%)		
1 year	63 (75.0%)	21 (25.0%)		
2 years	50 (57.5%)	37 (42.5%)		
3 years	40 (53.3%)	35 (46.7%)		
Two doses			12.947	0.012
1 month	79 (95.2%)	4 (4.8%)		
6 months	64 (82.1%)	14 (17.9%)		
1 year	68 (86.1%)	11 (13.9%)		
2 years	59 (79.7%)	15 (20.3%)		
3 years	45 (75.0%)	15 (25.0%)		
Stratified by time interval				
1 month			63.628	0.000
One dose	29 (34.9%)	54 (65.1%)		
2 doses	79 (95.2%)	4 (4.8%)		
6 months			4.706	0.030
One dose	55 (67.1%)	27 (32.9%)		
Two doses	64 (82.1%)	14 (17.9%)		
1 year			3.165	0.075
One dose	63 (75.0%)	21 (25.0%)		
Two doses	68 (86.1%)	11 (13.9%)		
2 years			9.060	0.003
One dose	50 (57.5%)	37 (42.5%)		
Two doses	59 (79.7%)	15 (20.3%)		
3 years			6.710	0.010
One dose	40 (53.3%)	35 (46.7%)		
Two doses	45 (75.0%)	15 (25.0%)		

**Table 2 vaccines-12-00290-t002:** GAMLSS construction and fitting.

Model	Distribution	μ	σ	υ	τ	AIC	SBC	Converged
M1	BCT	cs (0.066)	cs (−0.002)	cs (0.276)	cs (0.098)	1162.64	1256.48	Yes
M2	BCT	cs (0.066)	cs (−0.002)	cs (0.276)	pb (0.082)	1192.92	1357.78	Yes
M3	BCT	cs (0.066)	cs (−0.001)	pb (0.314)	pb (1.573)	1187.04	1347.66	Yes
M4	BCT	cs (0.065)	pb (−0.002)	pb (0.284)	pb (3.495)	1196.07	1366.65	Yes
M5	BCT	pb (−)	pb (−)	pb (−)	pb (−)	-	-	No
M6	BCT	pb (−)	cs (−)	pb (−)	pb (−)	-	-	No
M7	BCT	pb (−)	pb (−)	pb (−)	cs (−)	-	-	No
M8	BCT	pb (0.065)	pb (−0.001)	cs (0.279)	pb (1.076)	1191.51	1373.98	Yes
M9	BCT	pb (0.065)	pb (−0.001)	cs (0.279)	cs (0.485)	1155.83	1254.59	Yes
…	…	…	…	…	…	…	…	…
M10	BCCG	cs (0.066)	cs (−0.002)	cs (0.280)	-	1152.64	1223.02	Yes
M11	BCCG	cs (0.066)	cs (−0.002)	pb (0.281)	-	1152.78	1223.66	Yes
M12	BCCG	cs (0.065)	pb (−0.002)	pb (0.284)	-	1152.17	1219.78	Yes
M13	BCCG	pb (−)	pb (−)	pb (−)	-	-	-	No
M14	BCCG	pb (0.066)	cs (−0.001)	pb (0.287)	-	1146.7	1245.08	Yes
M15	BCCG	pb (0.066)	cs (−0.001)	cs (0.278)	-	1146.96	1227.08	Yes
M16	BCCG	pb (0.065)	pb (−0.001)	cs (0.282)		1145.83	1221.13	Yes
…	…	…	…	…	…	…	…	…

**Table 3 vaccines-12-00290-t003:** Centile reference values for VZV-IgG (logarithmic) in children aged 1–7 years.

Age	P_3_	P_10_	P_25_	P_50_	P_75_	P_90_	P_97_	P_2.5_–P_97.5_
1	1.393	1.515	1.667	1.882	2.170	2.532	3.056	1.378–3.144
2	1.508	1.688	1.900	2.181	2.524	2.897	3.343	1.487–3.409
3	1.430	1.661	1.920	2.236	2.585	2.928	3.295	1.401–3.346
4	1.456	1.798	2.138	2.509	2.875	3.200	3.518	1.411–3.560
5	1.519	1.831	2.145	2.491	2.835	3.143	3.446	1.477–3.486
6	1.524	1.774	2.039	2.348	2.672	2.975	3.285	1.492–3.327
7	1.523	1.758	2.015	2.321	2.649	2.964	3.293	1.493–3.338

## Data Availability

The data presented in this study are available on request from the corresponding author.

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
