# Peer review of "Study on Varicella-Zoster Virus Antibody Levels among Children Aged 1–7 Years in Changzhou, China"

_vaccines, 2024, doi:10.3390/vaccines12030290_

Round 1

Reviewer 1 Report

Comments and Suggestions for Authors

The manuscript by Wu, D and colleagues describe a cross-sectional VZV seroprevalence study in a relatively large cohort of VZV-VarV vaccinated young children in Changzhou (China). Compared to a 1 dose regimen, the 2-dose regiment resulted in a significantly higher VZV IgG seroprevalence. Based on these data, which resemble that of earlier studies in other countries, the authors conclude that a 2-dose VarV immunization strategy is recommended to protect young children from acquiring varicella. The study is of interest, the data presented largely support the conclusion and the manuscript is well-written. However, the manuscript contains several ethical and technical shortcomings that need to be addressed.

Major concerns:

1.      The questionnaire survey included multiple personal data that enables tracing back to the specific individual: how was data protection secured? Furthermore, provide the local IRB permit identifier number enabling review of its ethical status and clarify if the child itself or instead the child’s guardians provided their informed consent.

2.      Which vaccine was provided to the children: vOka based or subunit vaccine? Subsequent discussion of the data presented with that of earlier studies in relation to vaccine type is warranted. See also earlier comments on this manuscript that ended up in the Discussion section: first paragraph (lines 242-245).

3.      Provide details on the ELISA method used: which VZV antigen etc & product number. Why did the authors limited their study to detect VZV-specific IgG and not IgM?

4.      To acknowledge the benefit of VarV vaccination it is important to: (a) provide data on the seroprevalence of non-vaccinated children, especially among the 4 yr old children (national data are sufficient), (b) discuss why still about one-fourth children that are vaccinated 2-times remain VZV seronegative and how to improve this in the near future, (c) provide a reference of the study detailing the change in annual varicella incidence rate before and after introduction of the 2-dose VarV strategy in Changzhou (lines 253-255), and (d) high numbers of breakthrough cases reported in earlier 1 dose VarV studies indicated that a 2 dose regimen is mandatory to provide optimal protection. The authors acknowledge these studies, but also in their setting (lines 293-295) that warrants referring to the specific study. For both 4c and d, If no reference is available these data need to be included in this manuscript too.

Minor concerns:

1.      Clarify the household registration status: what is a ‘resident’ and ‘floating’ population?

2.      Provide the company or organization that delivered the statistical software.

3.      The data of the stratified analysis in Table 1 is difficult to interpret. Present data as new Figure.

4.      Title manuscript is unclear: (a) provide China as country were Changzhou is located and delete ‘an application of GAMLSS’ from the title. The latter is an abbreviation and not informative.

Comments on the Quality of English Language

No applicable.

Author Response

Thank you for your comments, see the word “Response to reviewer 1”for specific response.

Reviewer 2 Report

Comments and Suggestions for Authors

Varicella zoster virus (VZV) causes generalized herps zoster and becomes latent in peripheral neurons that upon reactivation can cause severely painful rash and neurological dysfunction. The virus is highly contagious, primarily transmitted respiratory droplets and especially problematic in schools with children particularly susceptible.  VZV vaccine regimens vary considerably world-wide, especially with respect to the number of doses used.

In this study, the efficacy of single and two-dose vaccination in the generation of an antibody response to VZV in children in Changzou, China was evaluated through a GAMLSS analysis.  The data show that, in the one-dose group, the seropositivity rate reached 75% at the one-year mark, then slowly declined, while, after a second dose, the positivity rate reached 95% after only one month post-vaccination, declined though 6 months postvaccination and subsequently increases until the one-year time point, before declining slowly.  

There are no discernible weaknesses in either the manuscript or in the study itself. The data are totally convincing, statistically significant and appropriately interpreted and present a definitive argument in support of the implementation of a two-dose vaccine regimen in children 4-7 years of age, preferably with a relatively short interval between vaccinations. 

Author Response

Thank you for your comments!

Reviewer 3 Report

Comments and Suggestions for Authors

            The authors have submitted a detailed analyses of antibody titers in children who have received varicella vaccination in one city in China.  The tables are complete. The data will be valuable as a guide to other public health officials as the assess whether to recommend varicella vaccination in their city or country. See 3 comments below.

1.     Section on Materials and Methods.  

Please add a new section called Vaccine.  Give the name of the vaccine used in this city and its manufacturer.  Is this varicella vaccine derived from the Oka varicella vaccine developed in Japan in the 1970s? Give the GenBank number if the Chinese vaccine product has been sequenced.

2.     Methods, section 2,3.  Give the product number of the ELISA kit purchased from VIRION/SERION company.

3.Discussion lines 294-297.  The varicella antibody titers after even 2 vaccinations in childhood will gradually decline by late adolescence. Please cite one more article written by F. Bianchi, et al.  Long term persistence of antibodies against varicella in fully immunized health care workers.  BMC Infectious Diseases, 25 May 2021.  PMID:  34034659. Please mention in the Discussion that varicella antibody titers likely will decline further if the titers are measured in young adults from China who were immunized during childhood.  In this study from Italy, only 67% of young adults still had detectable varicella antibody.

Comments on the Quality of English Language

Perhaps some minor adjustments. 

Round 2

Reviewer 1 Report

Comments and Suggestions for Authors

The authors approved the MS and addressed majority concerns. One important issue needs to be clarified:

Authors' response to comment 3 is not appropriate. The vaccine content and source is partly clarified in the M&M section 2.2 (lines 96-102). Three domestic vOka varicella vaccines were used and Genbank IDs of the respective seed virus stocks were provided. However, one of the 3 strains (AB097933.1) appears to be the parental, pathogenic Oka strain (pOka). This is really scary! 

Furthermore, it is not indicated if the vOka strains used in the 3 Chinese varicella vaccines are (a) all derived from the original Japanese vOka strain generated about 50 yrs ago and (b) if the genbank entries provided are obtained by sequencing the virus of a random vOka stock used to generate the vaccines? This is important, as we know that the vOka vaccine stocks can differ between companies, not only the vOka quasispecies but also specific mutations.

Author Response

Dear reviewer,

Firstly, to be honest, we konw less about the gene sequencing of the vaccine. And the Genbank numbers were mainly provided by the three vaccine companies after we communicated with them. After checking with Changchun BCHT Biotechnology company(also known as Baike) again, the staff of Bike told us that the Genbank number of the VarV produced by them was AB097932. We have made the appropriate changes, and we were told by staff at all three companies that the vaccine was derived from the original Japanese vOka strain, which was produced about 50 years ago. And according to them, the genbank entries provided were obtained by sequencing the virus of a random vOka stock used to generate the vaccines.

Kind regards

Round 3

Reviewer 1 Report

Comments and Suggestions for Authors

The authors improved the MS. A real pity the vOka strain info, used for the Chinese varicella vaccines info, could not be obtained from the companies. 

Author Response

Dear reviewer,

The Oka virus underwent attenuation through successive passages in cell culture. The resultant Oka strain vaccine, referred to as vOka-Biken, was administered to Japanese children in 1974. Following that, GlaxoSmithKline (GSK) Biologicals and Merck Sharp & Dohme (Merck & Co.) developed their own versions of the Oka vaccine. These were named Varilrix and Varivax, respectively, both originating from the Biken seed stock. In 1998, varicella vaccine became accessible in China. Like the Merck & Co. and GSK vaccines, all Chinese varicella vaccines were derived from the Biken seed virus.

Kind regards.